# Deep Learning-Based 3D Measurements with Near-Infrared Fringe Projection

**DOI:** 10.3390/s22176469

**Published:** 2022-08-27

**Authors:** Jinglei Wang, Yixuan Li, Yifan Ji, Jiaming Qian, Yuxuan Che, Chao Zuo, Qian Chen, Shijie Feng

**Affiliations:** 1Smart Computational Imaging Laboratory (SCILab), School of Electronic and Optical Engineering, Nanjing University of Science and Technology, Nanjing 210094, China; 2Smart Computational Imaging Research Institute (SCIRI), Nanjing University of Science and Technology, Nanjing 210019, China; 3Jiangsu Key Laboratory of Spectral Imaging and Intelligent Sense, Nanjing 210094, China

**Keywords:** fringe projection, speckle noise, phase retrieval, denoising, deep learning

## Abstract

Fringe projection profilometry (FPP) is widely applied to 3D measurements, owing to its advantages of high accuracy, non-contact, and full-field scanning. Compared with most FPP systems that project visible patterns, invisible fringe patterns in the spectra of near-infrared demonstrate fewer impacts on human eyes or on scenes where bright illumination may be avoided. However, the invisible patterns, which are generated by a near-infrared laser, are usually captured with severe speckle noise, resulting in 3D reconstructions of limited quality. To cope with this issue, we propose a deep learning-based framework that can remove the effect of the speckle noise and improve the precision of the 3D reconstruction. The framework consists of two deep neural networks where one learns to produce a clean fringe pattern and the other to obtain an accurate phase from the pattern. Compared with traditional denoising methods that depend on complex physical models, the proposed learning-based method is much faster. The experimental results show that the measurement accuracy can be increased effectively by the presented method.

## 1. Introduction

An optical three-dimensional (3D) measurement [1] is extensively used in many fields, such as industrial manufacturing, biomedicine, and defect detection, because of its high robustness, high efficiency, and high accuracy [2,3]. As a representative optical 3D measurement technique, fringe projection profilometry (FPP) is able to capture a full-field and high-resolution 3D image rapidly compared to the coordinate measuring machine that relies on physical contact [4,5,6]. The measured surface is illuminated with pre-designed fringe patterns and the phase is measured from the patterns and converted into 3D coordinates in FPP. Consequently, the accuracy of FPP is fundamentally dependent on the accuracy of phase demodulation. According to the used phase retrieval methods, classic methods such as Fourier transform profilometry (FTP) [7] and phase-shifting profilometry (PSP) are developed [8,9]. FTP that uses the filtering in the frequency domain can measure the phase through a single fringe pattern. However, it usually assumes that the surface under test is smooth and requires the spatial frequency of the projected grating to be sufficiently high. In contrast, PSP exploits the change in the light intensity of the pixels on the time axis to calculate the phase information of an object, thus showing a higher spatial resolution than FTP and making it suitable for phase measurements on complex surfaces.

Recently, the deep learning technique has been applied to 3D measurements, providing new potentials to improve the performance of phase recovery and 3D measurements [10,11,12,13,14,15,16,17,18,19,20,21,22]. Yan et al. [14] constructed a deep convolutional neural network (DCNN) consisting of 20 convolutional layers to process fringe image denoising. Jeon et al. [15] proposed a fast speckle noise reduction method for digital holograms using a multiscale CNN. Feng et al. [16,17] proposed a fringe analysis method based on deep learning, which can achieve a high-accuracy phase measurement and maintain the details of object contours. Qian et al. [18] proposed a deep learning-based geometric constraint and phase unwrapping method that can satisfy the measurement needs of a single absolute 3D shape. However, these methods are developed for visible fringe patterns, which may be compromised when these methods are used to handle invisible patterns of poor quality.

During the imaging of invisible infrared images, the unprocessed laser beam causes the uneven illumination of the detection area and creates a large amount of laser speckle in the detector image plane. The speckle noise causes a randomized distribution of image pixel amplitudes, producing a fuzzy, grainy distribution structure that blurs the fine features of the image. In a straightforward way, the speckle noise may be reduced by using phase-shifting methods with a large step. However, the efficiency would be decreased. Generally, speckle denoising methods based on image processing can be classified in the following categories: (1) spatial-domain denoising methods, (2) transform-domain denoising methods, and (3) learning-based denoising methods. Muhire et al. [23] applied the Wiener filter to the denoising of speckle images obtained from digital speckle interferometry. In the Wiener filter, a statistical estimate of the noise is obtained and minimized; however, it also causes blurring at sharp edges. Leng et al. [24] employed the Lee filter for the speckle denoising in the images reconstructed from digital holograms. By employing the criterion of minimum mean square error filtering, this filter achieves a great speckle denoising performance. Moreover, in the homogeneous regions, it denoises the speckles very well. However, it causes blurs at the edges and textures at the same time. Qian et al. [25,26] proposed the Fourier transform-based denoising method called windowed Fourier transform (WFT), which is a transform domain denoising method. An appropriate thresholding technique is applied to the obtained WFT coefficients of the speckled image in order to eliminate the spectral contribution of speckle noise. However, the threshold of this method needs to be determined by experience for different scenarios. Huang et al. [27] constructed another transform-domain denoising method known as bidimensional empirical mode decomposition (BEMD), which is the extension of the empirical mode decomposition. Without any thresholding function, it shows a great performance for speckle denoising, but it is computationally inefficient due to the use of the sifting process and the interpolation type in this algorithm. Zhang et al. [28] proposed a flexible denoising convolutional neural network, termed FFDNet. The FFDNet was proposed for the elimination of ordinary Gaussian noise and further implemented for speckle denoising by Hao et al. [29]. In addition, to perform image denoising, the block matching and 3D collaborative filtering (BM3D) method is widely applied [30,31,32], which is a fusion of spatial-domain denoising and frequency-domain denoising algorithms, and it can preserve the structure and details of images while ensuring the image of a good SNR. However, the BM3D algorithm is susceptible to the sigma parameter, which needs to be adjusted according to the input source to control the degree of denoising for different objects and environmental scenes. Further, its time cost is high, which may affect the measurement efficiency.

This paper proposes a 3D measurement method for near-infrared invisible fringe projection, which introduces the deep learning technique to eliminate the effect of speckle noise and produce accurate 3D models efficiently. Firstly, a deep learning denoising network is built to remove the speckle noise by learning the ground-truth results obtained by the BM3D. Then, another deep neural network is constructed to calculate the sine term and the cosine term of the phase with the filtered fringe pattern. The outputs of the phase retrieval network are substituted into the arctangent function for the final phase computation. The experiments demonstrate that our method can obtain high-precision phase information from a single fringe image with heavy speckle noise and achieve the 3D measurement accuracy of 80 μm.

## 2. Principles

In a typical setup of FPP, fringe patterns are projected on measured objects by a projector and then are captured by one or several cameras. The phase information is retrieved through fringe pattern analysis and then converted into 3D reconstructions. Our approach to NIR FPP consists of two parts: the NIR fringe pattern denoising and the phase measurement from the processed fringe image. As shown in Figure 1, we build a deep learning framework consisting of two convolutional neural networks (CNN1 and CNN2). CNN1 learns to remove the speckle noise in the raw fringe patterns, and CNN2 is trained to obtain the sine term (i.e., the numerator) and the cosine term (i.e., the denominator) of the phase from the processed fringe pattern, which is the output of CNN1. The wrapped phase can then be acquired by substituting these terms into the arctangent function. After the phase unwrapping and the stereo matching using the phase as the cue, the 3D model can be obtained.

### 2.1. The Elimination of Speckle Noise in NIR Fringe Pattern Using Deep Learning

Assuming that the reflected light intensity of the scenario is represented as *I*, the image impacted by speckle noise when captured by the camera can be computed as:(1)$$I^{\prime }=\delta \left(x,y\right)\times I,$$
where δ is multiplicative noise, and (x,y) is the pixel coordinate.

In order to remove the noise, the denoising algorithm we chose is BM3D:(2)$$I^{\prime }\overset{BM3D}{⟶}I^{B},$$
where $I^{B}$ is the denoised image.

The idea of BM3D is to use image block matching to collect and aggregate the similar structures and then orthogonally transform them to obtain a sparse representation, making full use of sparsity and structural similarity for filtering.

Although BM3D shows promising potentials for removing the speckle noise, it is complicated and time-consuming. In order to propose a flexible and efficient denoising strategy, we develop an end-to-end deep neural network for fringe pattern denoising. We construct pairs of training data from captured noisy images and clean images and then train a network to remove the noise from these given noisy images. The fringe images processed by BM3D are used as the ground-truth clean images. This ensures that the output of CNN1 enjoys accuracy and a satisfying denoising effect, which removes the noise while preserving the detail of image features. The structure of CNN1 is shown in Figure 2, which consists of a residual block and several convolutional layers. Here, *H* is the input image height and *W* is the image width. *C* is the number of filters which is set as 50 in our CNN1. To train the network, the loss function is defined as:(3)$${\mathrm{loss}}_{1}=\frac{1}{H\times W}{∥I_{g}-I^{B}∥}^{2},$$
where $I_{g}$ is the denoised image obtained by BM3D. $I^{B}$ is the output of CNN1.

### 2.2. Analysis of Denoised Fringe Pattern Using Deep Learning

The mathematical expression for the denoised fringe pattern processed by CNN1 can be written as:(4)$$I^{B}\left(x,y\right)=A\left(x,y\right)+B\left(x,y\right)\cos\phi \left(x,y\right),$$
where $I^{B}$ represents the intensity of the fringe pattern after the processing of CNN1, (x,y) is the pixel coordinate, *A* is the background light intensity, *B* is the fringe amplitude, ϕ is the desired phase distribution.

In most phase demodulation techniques, the background light intensity *A* is regarded as an interference term and should be removed. The wrapped phase map is recovered by applying an inverse trigonometric function to a fraction, whose numerator and denominator are the phase sine and the phase cosine, respectively:(5)$$\phi \left(x,y\right)=\arctan\frac{M(x,y)}{D(x,y)}=\arctan\frac{cB(x,y)\sin\phi (x,y)}{cB(x,y)\cos\phi (x,y)},$$
where *c* represents a constant dependent on the phase demodulation algorithm. The CNN2 is trained to predict the numerator M(x,y) and the denominator D(x,y) of the arctangent function by feeding the network with $I^{B}$.

The ground-truth data of CNN2 is generated by using FPP. In *N*-step phase-shifting algorithm, the fringe pattern can be written as:(6)$$I_{n}\left(x,y\right)=A\left(x,y\right)+B\left(x,y\right)\cos\left[\phi \left(x,y\right)-\delta_{n}\right],$$
where $I_{n}$ represents the *n*th captured image, the index n=0,1,…,N−1, and $\delta_{n}$ the phase shift that equals 2πn/N.

The object phase ϕ can be calculated using the least square method:(7)$$\phi \left(x,y\right)=\arctan\frac{\sum_{n=0}^{N-1}I_{n}\sin\delta_{n}}{\sum_{n=0}^{N-1}I_{n}\cos\delta_{n}},$$

Here, the phase information can be expressed as:(8)$$\phi \left(x,y\right)=\arctan\frac{M(x,y)}{D(x,y)},$$
where M(x,y) and D(x,y) are:(9)$$M\left(x,y\right)=\sum_{n=0}^{N-1}I_{n}\left(x,y\right)\sin\delta_{n}=\frac{N}{2}B\left(x,y\right)\sin\phi \left(x,y\right),$$
(10)$$D\left(x,y\right)=\sum_{n=0}^{N-1}I_{n}\left(x,y\right)\cos\delta_{n}=\frac{N}{2}B\left(x,y\right)\cos\phi \left(x,y\right),$$

The wrapped phase ϕ(x,y) has a truncated spatial distribution and 2π phase jumps. Therefore, the unwrapping process is required to obtain the absolute phase [6]. The absolute phase can be obtained by:(11)Φ(x,y)=ϕ(x,y)+2πk(x,y)
where Φ(x,y) is the absolute phase, k(x,y) represents the fringe order.

In this work, CNN2 is developed to learn M(x,y) and D(x,y), according to the denoised fringe pattern. Its structure is shown in Figure 3. The deep neural network is constructed by four path convolutional layers with residual blocks. From up to down, the sampling rate increases by the number of 2 with a rise in the depth of the path. By using this strategy, the network can extract both local and global features and finally combines them together to ensure the best performance of the network. The residual block can speed up the convergence of the deep network and improve its performance by adding layers with considerable depth. Moreover, the structure of the residual block can prevent overfitting while the network gets deeper. After different scales of downsampling, the tensors’ sizes are inconsistent. Therefore, upsampling blocks will be used to make the tenors from various paths uniform. The number of residual blocks per path is 4, and the number of filters (C) in the convolutional layer and the upsampling block is 50. For each path in the network, the tensor will be downsampled by 1, 1/2, 1/4, and 1/8 times, respectively, using different scales of pooling layers. In addition, to avoid the overfitting problem common to deep neural networks, L2 regularization is used in each convolutional layer of the residual and upsampling blocks, enhancing the network’s convergence ability. The loss function for CNN2 is described as:(12)$${\mathrm{loss}}_{2}=\frac{1}{H\times W}\left[{∥Y_{M}-G_{M}∥}^{2}+{∥Y_{D}-G_{D}∥}^{2}\right],$$
where $G_{M}$ and $G_{D}$ are the ground-truth numerator and denominator obtained by BM3D along with eight-step PS, respectively. $Y_{M}$ and $Y_{D}$ are the numerator and denominator predicted by CNN2, respectively.

## 3. Experiments

To verify the proposed method, an NIR FPP system was built, which consists of a MEMS-based single-axis infrared laser scanning module (1280 × 960 resolution) and two industrial cameras (acA640-750um, Basler). The wavelength of the NIR illumination is 830 nm. The cameras were equipped with two 5 mm lenses, in front of which we place two NIR band filters for capturing the desired NIR patterns. To collect the training data, we captured 800 fringe images of different scenes. The scene consists of many objects, most of which are plaster models. The raw NIR image is obtained by collecting images of different objects and their combinations at different angles. The BM3D was used to remove the speckle noise and generate the ground-truth data of CNN1. As different scenes require BM3D implementations of different parameters (e.g., the sigma), fine-tuning is thus needed when handling different scenes. To generate high-quality labels, we carefully tuned the parameter of the BM3D and found that speckle noise can be well removed when the proper sigma was selected from 6 to 14. To form the ground-truth data of CNN2, we applied the BM3D to the captured phase-shifting patterns. Then, the eight-step phase-shifting algorithm was used to calculate the numerator and the denominator by using the denoised patterns. To train the networks, 75% of the whole dataset was used for training and the remaining 25% for validation. Before being fed into the two networks, the input image was divided by 255 for normalization. The adaptive moment estimation (ADAM) was used to tune the parameters to find the minimum of the loss function for CNN1 and CNN2. All the DNN’s training and testing were implemented in Python by using Keras by using an NVIDIA GeForce GTX 1080 Ti GPU with 11 GB video memory. The total training time for CNN1 and CNN2 is 6 and 10 h, respectively. Their loss curves are shown in Figure 4. For CNN1, as shown in Figure 4a, we can see that the loss curves of the training data and validation data converge, and their final loss values are around 3. For CNN2, as shown in Figure 4b, the loss curves converge to values near 6.

To test the proposed CNN1, we measured some objects that were not in the testing set. The results are shown in Figure 5 and Figure 6. In Figure 5, the first column is the original NIR fringe images captured by the camera. We can see that these images were heavily destroyed by the laser speckle noise. The second column shows the NIR fringe images processed by the BM3D, which are treated as the ground truth. The last column shows the output of our CNN1, which is comparable to ground truth in terms of the noise removal and the preservation of the edge details. We plotted the 300th row of the third scene, and the results are shown in Figure 6. We can clearly see that CNN1 can remove the noise effectively. Moreover, we also tested the efficiency of the proposed CNN1. As shown in Table 1, the time cost of the BM3D is around 1.9 s. When CNN1 was applied, the processing time could be reduced to less than 0.07 s, showing that the proposed CNN1 is 30 times faster than the BM3D in removing the speckle noise.

The CNN2 was then trained with the denoised fringe patterns obtained by CNN1. Figure 7 shows the predicted results of CNN2. The first and second columns of Figure 7 show the numerator and denominator obtained by CNN2, respectively. The third column shows the wrapped phase calculated by Equation (Equation 5), and the fourth column shows the absolute phase obtained by the temporal phase unwrapping. To test the accuracy of the predicted results of CNN2, the absolute phase of CNN2 was compared with the ones obtained with (1) the raw fringe images analyzed by the three-step phase-shifting algorithm and (2) the denoised fringe images with the three-step phase-shifting algorithm.

The results of the phase error are shown in Figure 8. From left to right, the first column shows the ground-truth absolute phase obtained from the NIR fringe image after BM3D processing with the eight-step phase-shifting algorithm. The second column is the absolute phase of the original NIR fringe image obtained by the three-step phase-shifting algorithm. The mean absolute error (MAE) is 0.0716, 0.1071, and 0.1117 rad for the scenes, respectively. The third column shows the absolute phase of the NIR fringe image, which is obtained by BM3D processing first and the three-step phase-shifting algorithm later. The corresponding error is 0.0571, 0.0760, and 0.0710 rad. The last column shows the absolute phase obtained by our method and the error is 0.0278, 0.0356, and 0.0332 rad. It can be seen that our method can effectively reduce the error caused by the speckle noise in NIR fringe projection.

When two views of the fringe patterns were processed by our method, the 3D reconstructions were performed, and the results are shown in Figure 9. From the first to the last column, we show the 3D reconstruction results of the ground-truth method, the raw NIR patterns followed by the three-step phase-shifting algorithm, the NIR fringes denoised by BM3D followed by the three-step phase-shifting algorithm, and the proposed method, respectively. For the results of the second column, the objects were reconstructed as noisy models and there are even some spike errors there. For the results of the third column, we can see that the noise error has been removed to some extent because of the merit of the BM3D. However, there are still some spike errors. From the results of our method, we can see that most of the noise error has been eliminated, which shows that our results are comparable to the ground-truth labels.

In addition, to quantitatively estimate the reconstruction accuracy of our approach, we measured a ceramic sphere, whose radius, measured by a coordinate measuring machine, is 25.4 mm. The results are shown in Figure 10. The reconstructed ceramic sphere of our method has a radius of 25.369 mm with an error of 31 μm and the RMS error of 80 μm. Moreover, the surface of the sphere obtained by our method is smoother than the one obtained by the raw fringe pattern with the three-step phase-shifting algorithm. We can see that the result of the proposed method is also superior to the one obtained by the fringe images denoised by BM3D together with the three-step phase-shifting algorithm, as a smaller RMS error and the smoother shape are observed. In addition, our method only used a single fringe image to retrieve the wrapped phase, which also shows higher efficiency.

## 4. Conclusions

In this paper, we have proposed a deep learning-based framework that can decrease the influences of speckle noise in the image captured by near-infrared laser and enhance the quality of 3D reconstruction results. The framework consists of two deep neural networks performing different tasks: one is the image denoising network, which is responsible for the denoising of the fringe pattern, and the other is the phase retrieval network, which obtains the accurate phase from the pattern. Inspired by the approaches to removing the noise in general visual images, we have developed the proposed method. We believe it has the potential to be extended to more kinds of images. Our method achieves an accuracy of 80 μm using only one single fringe image. The experimental results have shown that compared with traditional denoising methods, such as BM3D, the proposed learning-based method can increase the denoising speed by more than an order of magnitude. Moreover, the accuracy of the 3D reconstruction has been improved effectively with our method.

## Figures and Tables

**Figure 1 sensors-22-06469-f001:**
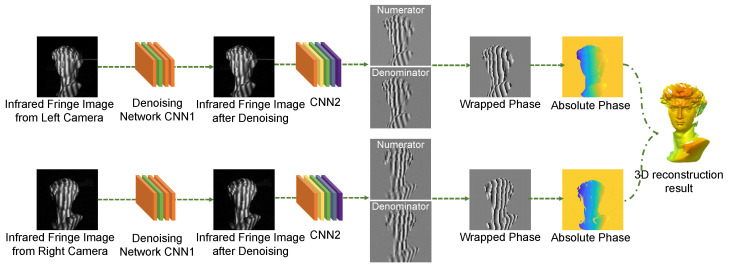
The flowchart of the proposed deep learning-based 3D measurements using NIR FPP. For CNN1, the input is the raw fringe image with speckle noise and the output is the denoised image. For CNN2, it learns to obtain the numerator and denominator. As the phase can be used as temporary texture, the 3D reconstruction is then calculated with stereo vision.

**Figure 2 sensors-22-06469-f002:**
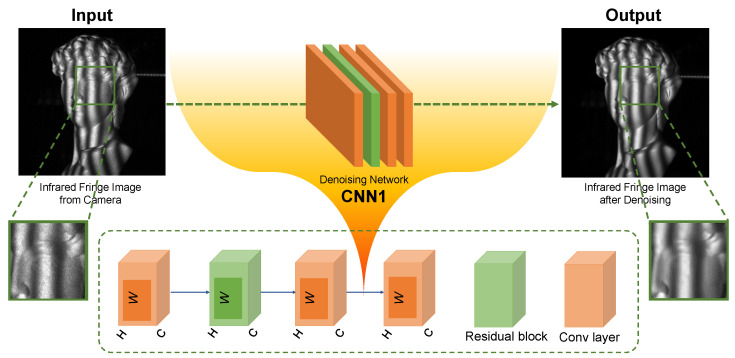
Schematic diagram of the denoising network CNN1, consisting of a convolutional layer and multiple residual blocks.

**Figure 3 sensors-22-06469-f003:**
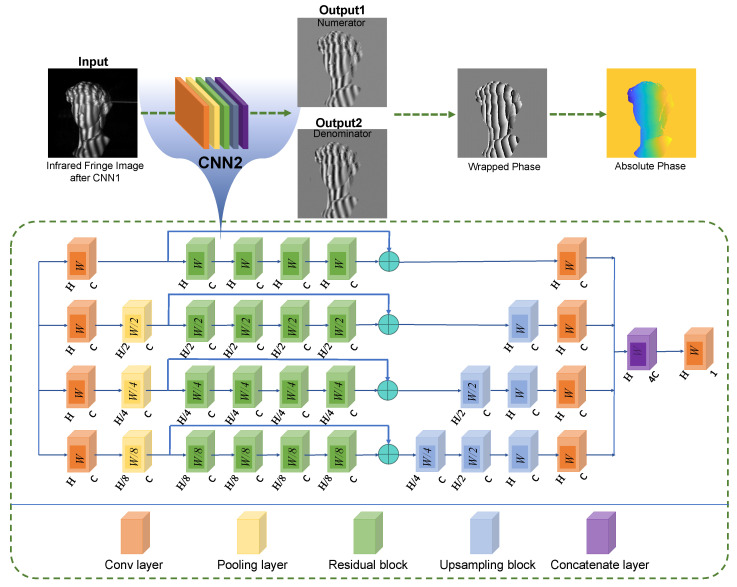
Schematic representation of phase information in fringe images demodulated using deep neural network CNN2.

**Figure 4 sensors-22-06469-f004:**
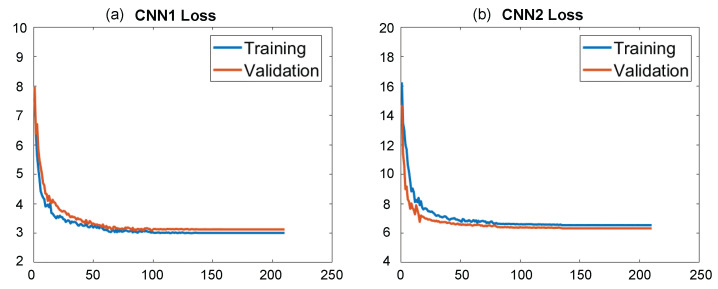
The loss curve of (**a**) CNN1, (**b**) CNN2.

**Figure 5 sensors-22-06469-f005:**
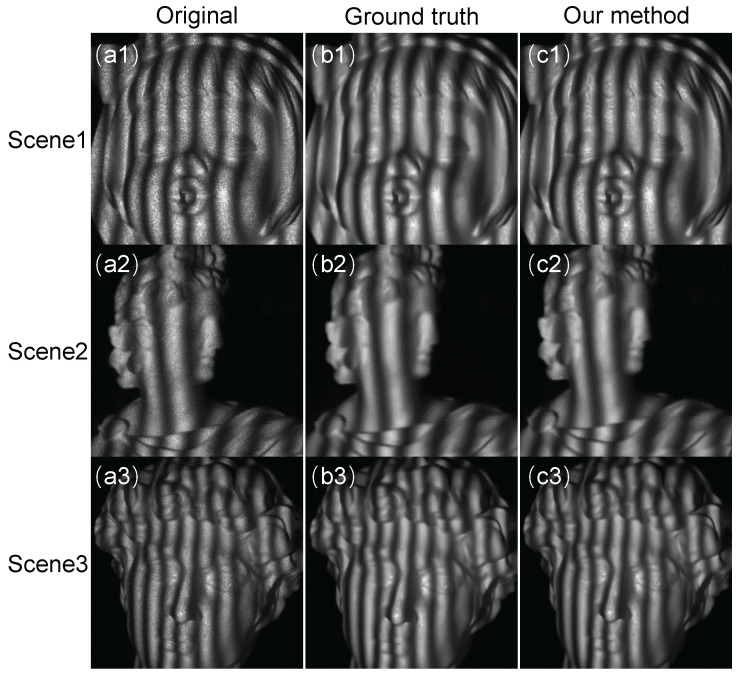
The performance of the trained CNN1. (**a1**–**a3**) The captured raw NIR fringe patterns of different scenes. (**b1**–**b3**) The ground-truth-filtered NIR fringe patterns processed by BM3D. (**c1**–**c3**) The filtered NIR fringe patterns obtained by CNN1.

**Figure 6 sensors-22-06469-f006:**
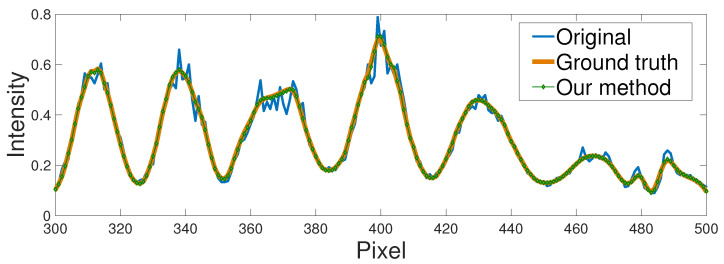
The comparison of the algorithm in the 300th row from Figure 5a3,b3,c3.

**Figure 7 sensors-22-06469-f007:**
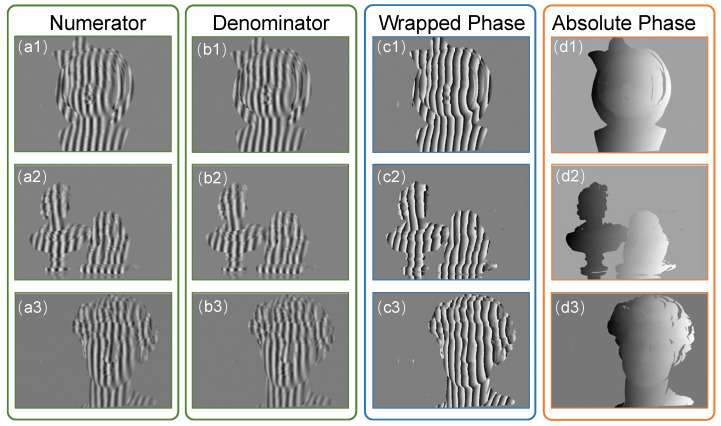
The numerator (**a1**–**a3**) and denominator (**b1**–**b3**) estimated by our method. (**c1**–**c3**) The wrapped phase calculated with numerator and denominator. (**d1**–**d3**) The absolute phase obtained by TPU using the wrapped phase.

**Figure 8 sensors-22-06469-f008:**
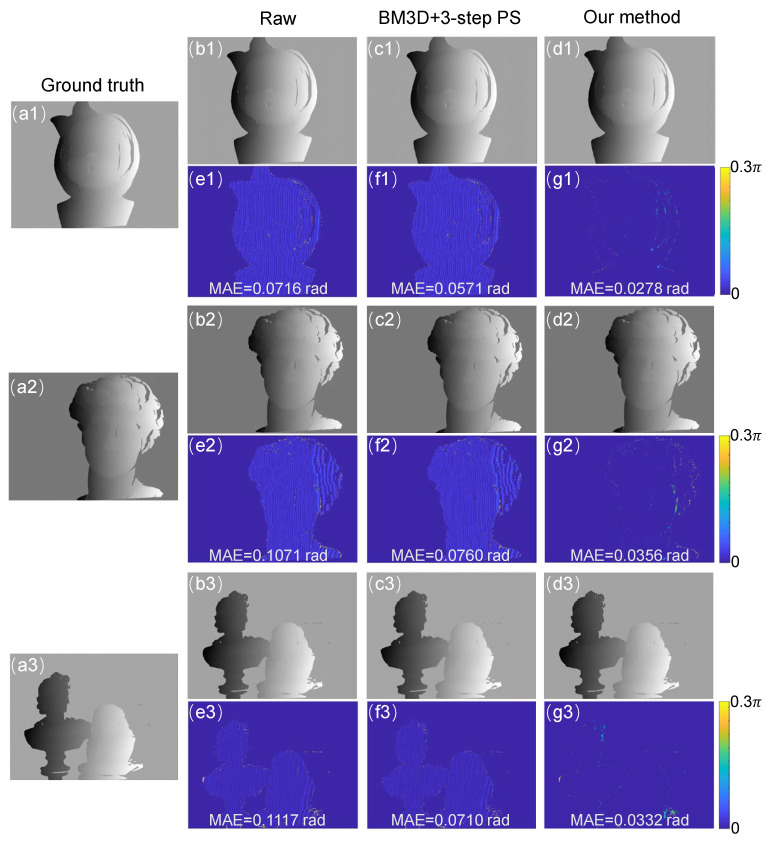
(**a1**–**a3**): The ground-truth label of the unwrapped phase which was calculated by the NIR fringes denoised by BM3D followed by the eight-step phase-shifting algorithm. The unwrapped phase obtained by (**b1**–**b3**) the raw NIR patterns followed by the three-step phase-shifting algorithm, (**c1**–**c3**) the NIR fringes denoised by BM3D followed by the three-step phase-shifting algorithm, and (**d1**–**d3**) our method. (**e1**–**e3**,**f1**–**f3**,**g1**–**g3**): Absolute phase error maps of the corresponding cases.

**Figure 9 sensors-22-06469-f009:**
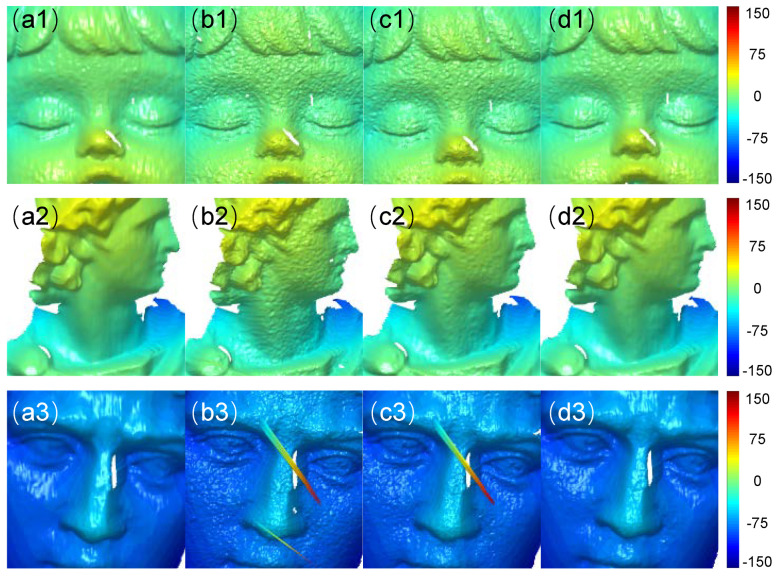
The 3D reconstruction of the NIR fringes obtained by (**a1**–**a3**) BM3D denoising followed by eight-step phase-shifting algorithm, (**b1**–**b3**) three-step phase-shifting algorithm, (**c1**–**c3**) BM3D denoising followed by three-step phase-shifting algorithm, and (**d1**–**d3**) our method.

**Figure 10 sensors-22-06469-f010:**
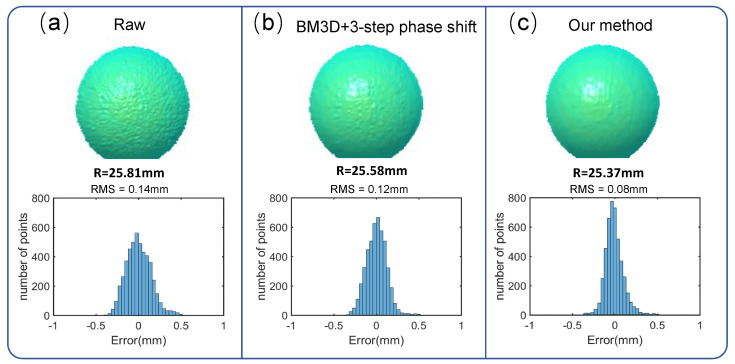
The 3D reconstructed sphere (top) and error pixels distribution (bottom) obtained by (**a**) direct three-step PS of the original IR fringes, (**b**) BM3D denoising with three-step PS, and (**c**) our method.

**Table 1 sensors-22-06469-t001:** Comparison of image denoising processing time between BM3D and our deep learning-based method for different scenes.

Time Cost of Fringe Analysis	BM3D/s	Our Method/s
Scene 1	1.983	0.0648
Scene 2	1.995	0.0673
Scene 3	1.997	0.0633

## Data Availability

Not applicable.

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
