# Peer review of "Deep Learning-Based 3D Measurements with Near-Infrared Fringe Projection"

_sensors, 2022, doi:10.3390/s22176469_

Round 1

Reviewer 1 Report

This manuscript proposes a deep learning-based framework consisting of two deep neural networks to perform fringe image denoising and phase analysis, respectively. This method can reduce the influence of speckle noise in near-infrared laser-captured images and improve the 3D reconstruction. Overall, the idea is practical and the work shows potential to handle the noise in the low-cost structured light 3D measurement systems using the MEMS-based projection modules.

       I have some short questions: (1) Why did you choose to use the BM3D method to obtain the denoising fringe image as the labels of the network? How to use the denoising method to generate high quality labels? (2) Please list the detailed parameter information in the network model training process and supplement them in your manuscript, such as training time, prediction time, training loss value, validation loss value of the model, the loss curves, etc. (3) In Figure 8, the author should give the colorbar of the images. (4) Could the authors provide more information on how to collected the training data in their experiments? (5) Can the method be extended to general image denoising and not limited to IR fringe image denoising?

Reviewer 2 Report

The authors present an interesting work regarding the improvement of near-infrared (NIR) 3D fringe projection 3D scanning employing deep-learning strategies. The main objective is to reduce the usual speckle noise characteristic of NIR images and thus increase the 3D reconstruction quality and potential for measurements within. The approach to the problem was to use two convolutional neural networks, one for learning how to remove the speckle noise and the other to reconstruct the phase, for each of the cameras, and then reconstruct the 3D image from both results. Although not presenting a high level of novelty, there’s enough of it to be of interest to readers.

In general, the paper is well organized and uses a clear language and figures to complement the text. The introduction adequately explains the problem and the state of the art, while the principles that are used in the author’s approach are explained in a dedicated section. Since these principles are mainly known from the deep-learning field, it would be nice to have references “here and there”, so the reader can deepen some of the concepts, if he wants to. This is just a suggestion, but, as I said, it would improve the usefulness of this section.

The experimental part is well explained, although joining the methodologies with the results and analysis in the same section (without even sub-sections to separate both) isn’t the most common choice, in the particular case of this paper, readability through this section is (surprisingly) fluid. Conclusions adequately resume the work and outcomes of it.

Thus, the paper has potential interest to readers of Sensors. Only a few minor points that should be addressed:

1 – In line 197, the radius of the sphere is presented as being 25.3689 mm with an error of 31.148 um and RMS error of 80.175 um. Why presenting the error with such precision? Considering the radius, it seems mor adequate to have the errors as 31 um and 80 um, respectively.

2 – In Figure 9, besides the same comment regarding the number of decimal cases (the RMS should be presented with one, or at most, two decimal cases), the legend can be improved. First, there’s an unnecessary repetition of words (e.g., it could be: “The 3D reconstructed sphere by (a) direct three-step PS of the original IR fringes, (b) BM3D denoising with three-step PS, and (c) our method”). Second, it should be clearly identified what is presented on top and bellow in each case (e.g. “The 3D reconstructed sphere (top) and error pixels distribution (bottom) by (…)”).

This comment can be extended to figures 7 and 8.

3 – Still in Figure 9, having the vertical axis equal on all the three cases (case c varies only between 0 and 600 pixels) would help readability.
